# Three years of soil moisture observations by a dense cosmic-ray neutron sensing cluster at an agricultural research site in north-east Germany

Maik Heistermann[1], Till Francke[1], Lena Scheiffele[1], Katya Dimitrova Petrova[1], Christian Budach[1], Martin Schrön[2], Benjamin Trost[3], Daniel Rasche[4,1], Andreas Güntner[4,1], Veronika Döpper[5,6], Michael Förster[5], Markus Köhli[7,8], Lisa Angermann[1], Nikolaos Antonoglou[9,10], Manuela Zude-Sasse[3], and Sascha E. Oswald[1]

[1]Institute of Environmental Science and Geography, University of Potsdam, Karl-Liebknecht-Straße 24–25, 14476 Potsdam, Germany
[2]UFZ - Helmholtz Centre for Environmental Research GmbH, Dep. Monitoring and Exploration Technologies, Permoserstr. 15, 04318, Leipzig, Germany
[3]Leibniz-Institut für Agrartechnik und Bioökonomie (ATB), Max-Eyth-Allee 100, 14469 Potsdam, Germany
[4]GFZ German Research Centre for Geosciences, Section Hydrology, Telegrafenberg, 14473 Potsdam, Germany
[5]Technical University of Berlin, Geoinformation in Environmental Planning Lab, Straße des 17. Juni 135, 10623 Berlin, Germany
[6]now at Alfred Wegener Institute (AWI), Helmholtz Centre for Polar and Marine Research, Research Unit Potsdam, Telegrafenberg A43, 14773 Potsdam, Germany
[7]Physikalisches Institut, Heidelberg University, Im Neuenheimer Feld 226, 69120 Heidelberg, Germany
[8]Physikalisches Institut, University of Bonn, Nussallee 12, 53115 Bonn, Germany
[9]GFZ German Research Centre for Geosciences, Sect. Space Geodetic Techniques, Telegrafenberg, 14473 Potsdam, Germany
[10]Institute of Geosciences, University of Potsdam, Karl-Liebknecht-Straße 24–25, 14476 Potsdam, Germany

**Correspondence:** Maik Heistermann (maik.heistermann@uni-potsdam.de)

**Abstract.** Cosmic-ray neutron sensing (CRNS) allows for the estimation of root-zone soil water content (SWC) at the scale of several hectares. In this paper, we present the data recorded by a dense CRNS network operated from 2019 to 2022 at an agricultural research site in Marquardt, Germany - the first multi-year CRNS cluster. Consisting, at its core, of eight permanently installed CRNS sensors, the cluster was supplemented by a wealth of complementary measurements: data from seven additional temporary CRNS sensors, partly collocated with the permanent ones, 27 SWC-profiles (mostly permanent), two groundwater observation wells, meteorological records, and global navigation satellite system reflectometry (GNSS-R). Complementary to these continuous measurements, numerous campaign-based activities provided data by mobile CRNS-roving, hyperspectral imagery via unmanned aerial systems (UAS), intensive manual sampling of soil properties (SWC, bulk density, organic matter, texture, soil hydraulic properties), and observations of biomass and snow (cover, depth, and density). The unique temporal coverage of three years entails a broad spectrum of hydro-meteorological conditions, including exceptional drought periods, extreme rainfall, but also episodes of snow coverage, as well as a dedicated irrigation experiment. Apart from serving to advance CRNS-related retrieval methods, this data set is expected to be useful for various disciplines, e.g. soil and groundwater hydrology, agriculture, or remote sensing. Hence, we show exemplary features of the data set in order to highlight the poten-





# 1 Introduction

## 1.1 Towards closing the scale gap in soil moisture observation

A large body of literature highlights the significance of soil moisture as a key state variable of the earth system. Yet, soil moisture appears to elude our attempts to obtain representative observations: point-based measurements lack both representativeness and coverage (Blöschl and Grayson, 2000) while remote sensing struggles with issues such as small penetration depth and low overpass frequencies (Peng et al., 2021). Since Zreda et al. (2008), cosmic-ray neutron sensing (CRNS) has emerged as a promising option to address these issues, and hence to close the scale-gap between point measurements and large-scale soil moisture retrievals. The advantage of the CRNS sensor is its considerable horizontal (100–200 m) and vertical (20–50 cm) footprint (Schrön et al., 2017). It is hence considered to efficiently average across small-scale heterogeneity, allowing to obtain continuous estimates of "root-zone" soil moisture at the scale of several hectares (see e.g. Zreda et al., 2008; Desilets et al., 2010; Köhli et al., 2020, for further details). To that end, soil moisture is estimated from the epithermal neutron intensity by means of a conversion function. This typically involves the calibration of one parameter, $N_0$, on the basis of a sufficient number of soil moisture measurements in the sensor footprint (Schrön et al., 2017; Köhli et al., 2020).

Furthermore, different application scenarios of the CRNS technology have been developed in order to obtain spatial soil moisture estimates beyond the isolated footprint of a single, stationary CRNS sensor. The application of CRNS roving, for instance, involves a mobile neutron detector that is moved within a study area (Desilets et al., 2010; Schrön et al., 2018). That way, the spatial distribution of soil moisture along the roving transect can be inferred in terms of a snapshot in time, given that potential sources of bias (e.g. from road material or biomass effects) are sufficiently addressed (Fersch et al., 2018; Schrön et al., 2021).

As an alternative, dense clusters of stationary CRNS were proposed as an option to retrieve the spatial *and temporal* distribution of soil moisture. While also employing a number of detectors, they differ from CRNS networks (e.g. COSMOS, COSMOS UK, COSMOS Europe, see Zreda et al., 2012; Evans et al., 2016; Bogena et al., 2022) by having adjacent or even overlapping footprints. So far, data from two campaigns have been published which implemented dense clusters for a period of two months: the first, in the early summer of 2019, took place in a pre-alpine, mainly pastoral headwater catchment in southern Germany where 24 CRNS sensors were placed within an area of 1 km$^2$ (Fersch et al., 2020; Heistermann et al., 2021). The second, in the autumn of 2020, took place in the Eifel mountains, western Germany, where 15 CRNS were operated in the forested 0.4 km$^2$ Wüstebach catchment for three months (Heistermann et al., 2022a).



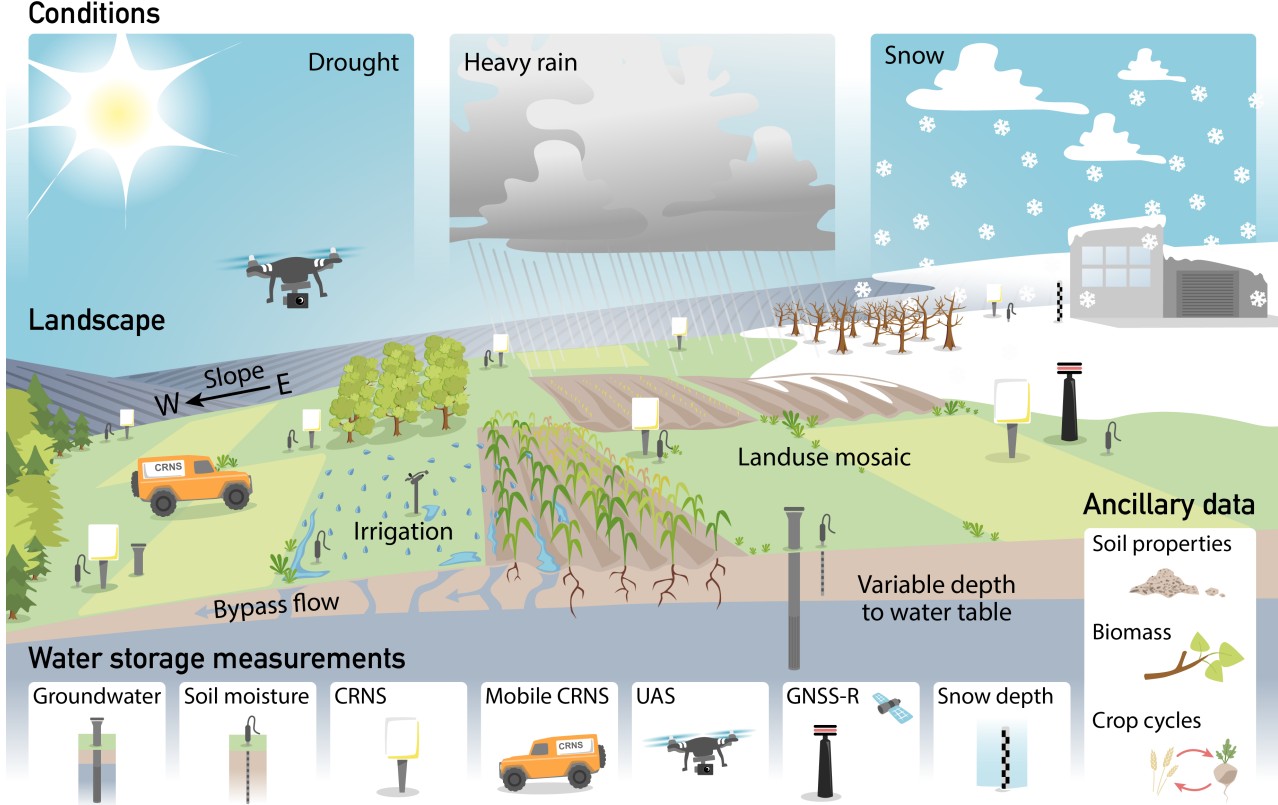

**Figure 1.** Graphical overview of the Marquardt Cluster, including an schematic view of the landscape, applied measurement techniques, and notable weather conditions over the study period.

## 1.2 Three years of dense CRNS observations: the Marquardt Cluster

In August 2019, the research unit *Cosmic Sense*, funded by the German Research Foundation (DFG), launched a dense CRNS
cluster at an agricultural research site in Marquardt (about 10 km northwest of Potsdam, Germany) - the first of its kind designed
for long-term operation. This effort will be referred to as the "Marquardt Cluster"(MqC). As of today, MqC is still operational,
and its core consists of eight CRNS sensors operated in an area of around 10 ha. As in the CRNS clusters previously presented
by Fersch et al. (2020) and Heistermann et al. (2022a), MqC was integrated in existing research infrastructure, here provided by
the Leibniz Institute for Agricultural Engineering and Bioeconomy (ATB, https://www.atb-potsdam.de), and complemented by
various observations, e.g. conventional reference measurements of soil moisture ("ground truth"), mapping of soil and biomass,
groundwater observations, CRNS roving, as well as alternative soil moisture retrieval techniques based on UAS-based remote
sensing and GNSS-R (Global Navigation Satellite Systems reflectometry). The underlying motivation of the dense CRNS
cluster was, on the one hand, to obtain representative soil moisture estimates for an area beyond the footprint of a single CRNS

sensor, and, on the other hand, to obtain information about the spatial variability of soil moisture at a length scale that is smaller

than the diameter of a single CRNS footprint.

Figure 1 provides a graphical overview of the MqC. There are various features which are specific to this endeavour, and which make it unique in comparison to both previous CRNS clusters, hence allow for new research opportunities:

– With a density of 8 CRNS sensors per 10 ha, MqC features a significant leap in density compared to previous clusters (3 and 2.4 sensors per 10 ha in Heistermann et al. (2022a) and Fersch et al. (2020), respectively), and hence a considerable

overlap of footprints; this aims for an improved identification of heterogeneity at the sub-footprint scale.

– The two aforementioned clusters span a few months each; the MqC is the first dense CRNS cluster operated over several years, spanning all seasons and allowing to observe more diverse conditions and processes, including periods of drought and snow cover as well as a set of heavy rainfall events at different durations.

– Focus on an agro-ecosystem with very diverse management: the site comprises traditional field crops, meadows, biomass

crops and orchards, irrigated as well as rainfed management, and adjacent forests.

– A large number of vertical soil moisture profile probes not only allows to study the effect of the vertical soil moisture distribution on the CRNS signal, but also, together with groundwater observations, to investigate the processes of infiltration and groundwater recharge.

– The data set includes a dedicated irrigation experiment: a selected plot was intensively irrigated while being monitored

with a cross-scale combination of sensors, including hyperspectral UAS-based remote sensing and CRNS roving.

– On-site muon-monitoring allows to study novel methods for the local correction of the incoming neutron flux.

– MqC is a typical lowland site in the transition from maritime to continental climate, thus representing a landscape typical for large parts of northern Central Europe.

## 1.3   Structure of this paper

This paper presents the MqC data acquired between August 2019 and November 2022. The study area is introduced in Sect. 2; the acquisition and, partly, the processing, of different subsets of the data is documented in Sect. 3. Other relevant data from third parties are addressed in Sect. 4. In Sect. 5, we highlight selected features of the obtained data with regard to various event types (snow, drought, heavy rainfall, irrigation) while the conclusions in Sect. 6 outline research perspectives with regard to the published data set.

## 2   Study site


The study area in Marquardt is a lowland agricultural site in the north-east of Germany. Various practical features made it a suitable candidate location for MqC, e.g. the vicinity to the participating institutions, the integration in an existing research



context, and the security provided by a complete fencing of the perimeter. The diversity of agricultural land use and management types, together with its high-level documentation, was a desirable feature with the regard to the investigation of a
heterogeneous landscape.

**Figure 2.** Overall setup of the MqC cluster at the ATB research site in Marquardt (permanent and temporary CRNS, SWC profiles, groundwater observations, tensiometer, GNSS antenna, climate station), and calibration sampling points from November 2019 and October 2022; the shown orchard plots (blueish green) are drip-irrigated; the irrigation experiment (red outline) was performed with a sprinkler; OSM layers were used to represent buildings and forested areas (© OpenStreetMap contributors, 2023, distribution under ODbL license). Inset map: Location of MqC in Germany and the two previous (short-term) CRNS clusters (JFC-2019 & JFC-2020, see Heistermann et al., 2022a).

On-site meteorological monitoring is available since 2009. The closest long-term climate station is located in Potsdam (at 12 km distance), operated by the German national meteorological service (Deutscher Wetterdienst, DWD hereafter). At this





DWD station, the average annual precipitation is 584 mm (1981–2010), the average annual temperature is 9.3 °C (Cfb according to Köppen's climate classification). The Marquardt site is located at about 40 m a.s.l. It sits on a gentle hillslope sloping

westwards, with distances to the unconfined groundwater table ranging from around 1.5 to 10 m. The soils are dominated by periglacial sand deposits over glacial till. The sand content ranges between 68 and 91 %, silt content is 8–27 %, and clay 0.6–4.4 %. The soil organic matter content ranges between 0.4 and 17.3 %.

The Marquardt site comprises approx. 20 field plots, which host annual and permanent crops. In the observation period of 2019–2022, the core area covered by the MqC was dominated by orchards (cherry, apple), field crops (cereals, alfalfa, sugar

beets, maize), meadows and a biomass plantation stocked with young poplars. Details are contained in the data repository.

**Table 1.** Overview of section 3: brief summary of each data subset, main observed variables and units, temporal coverage. Specific details can be found in the subsections and the json files which document each data subset in the repository (see also Tab.4).

| Sect. | Data subset | Main observation variables (units) | Temporal coverage |
|---|---|---|---|
| 3.2 | Stationary CRNS sensors (8 permanent, 7 temporary) recorded epithermal neutron counts, and meteorol. variables | Neutron counts, air pressure (hPa) and temperature (°C), rel. humidity (%) | Aug 2019-Nov 2022 |
| 3.3 | One muon detector to represent temporal variability of incoming fast neutrons | Muon counts, air pressure (hPa) and temperature (°C), rel. humidity (%) | May 2021-May 2022 |
| 3.4 | Roving CRNS tracks across the entire MqC site, with car and handwagon, at 4 dates as part of the irrigation experiment | Neutron counts, air pressure (hPa) and temperature (°C), rel. humidity (%) | Jul-Aug 2020 (four days) |
| 3.5 | GNSS-R antenna recorded signal-to-noise ratio (SNR) as well as ID and relative position of GPS satellites | SNR (db-Hz), satellite ID, elevation and azimuth (rad) | Jan 2019-Nov 2022 |
| 3.6 | UAS-based acquisition of hyperspectral imagery at 4 dates, coordinated with the irrigation experiment | Reflectance (-), SWC ($m^3/m^3$) derived from reflectance | Aug 2020 (four days) |
| 3.7 | Leaf area index measured at 4 dates (Plant Canopy Analyzer) | Leaf area index (-) | Jul-Aug 2020 |
| 3.8 | SWC time series at 27 profiles (dielectric measurements) | Permittivity (-), SWC ($m^3/m^3$) | Oct 2019-Nov 2022 |
| 3.9 | Groundwater level recorded at 2 observation wells | Distance of GW head to surface (m) | Aug 2019-Nov 2023 |
| 3.10 | Matric potential at 3 profiles down to 200 cm (tensiometers) | Matric potential (kPa) | Jun 2020-Nov 2022 |
| 3.11 | Campaigns on 2 dates with manual soil sampling of the upper 30 cm of the soil (split-tubes, ThetaProbes, lab analysis) | Permittivity, SWC ($m^3/m^3$), bulk density (g/cm$^3$), SOM (g/g), texture | Nov 2019, Oct 2022 |
| 3.12 | Lab evaporation experiment with kupF device to determine soil hydraulic properties | Matric potential (kPa), SWC ($m^3/m^3$), Ksat (cm/min) | Aug-Nov 2020 |
| 3.13 | Snow height monitoring and snow sampling (various techniques), areal imagery mosaics from 2 UAS overflights | Snow height (cm) and density (g/cm$^3$), precipitation (mm), RGB images | Feb 2021 |
| 3.14 | Landuse, crop cycles, aboveground dry biomass (AGB) | AGB (kg/m$^2$), harvest dates | Aug 2019-Sep 2022 |
| 3.15 | Irrigation experiment on one grassland plot with 3 irrigation phases, coordinated with CRNS roving & remote sensing | Spatial setup, irrigated amount (mm), SWC ($m^3/m^3$) | Jul-Aug 2020 |





## 3   Methods and data

### 3.1   Overview

This section documents the data acquired so far at the Marquardt site. The core of the data set consists of continuous time series of eight CRNS sensors (section 3.2), together with a set of 27 vertical SWC profiles (section 3.8). A comprehensive overview
of the various data subsets is provided in Tab. 1.

### 3.2   Stationary CRNS data

Between August 2019 and November 2022, eight stationary CRNS were operated as core component of the MqC. Sensors from different manufacturers were included: Four devices were manufactured by Hydroinnova LLC (Albuquerque, USA), three by Quaesta Instruments LLC (Tucson, USA), and one by Lab-C LLC (Sheridan, USA), now linked to Quaesta Instruments.
Seven additional sensors were operated for limited time periods, including devices from StyX Neutronica GmbH (Mannheim, Germany) and Finapp S.r.l. (San Pietro in Cariano, Italy). Table 2 provides an overview of the detectors and their sensitivities relative to a reference device (see Heistermann et al., 2022a, for details). Dividing by the sensitivity, neutron count rates observed by different sensors become comparable. As for neutron detection, most of the sensors are based on detector gases such as $^3$He gas (CRS-1000, CRS-2000) or $^{10}$BF$_3$ enriched gas (CRS-1000-B, CRS-2000-B, B-E1-4). "Hydro Sense dual"
uses a multiwire proportional chamber with solid $^6$Li (Fersch et al., 2020; Patrignani et al., 2021); StX-140-5-15 apply $^{10}$B-lined converters; the FINAPP3 probe relies on a multi-layer zinc sulfide and Ag-100 doped scintillator mixed with $^6$Li fluoride powder (Gianessi et al., 2022). In addition to epithermal neutrons, two devices also recorded thermal neutron count rates which might have the potential to support the separation of signals from soil moisture, vegetation, or snow (Tian et al., 2016; Jakobi et al., 2018). Further recorded variables include relative humidity, air temperature, and barometric pressure.
115       The locations of the CRNS sensors are shown in Fig. 2. Sensor placement was guided by various criteria: (i) to create significant overlap in the core area of the cluster, (ii) to cover the site along the hillslope gradient, (iii) to place some sensors close to the irrigated orchards, (iv) to minimize interference with agricultural management operations (sensors could not be placed on cropped fields).

The prime observation variable of the CRNS sensors are count rates of detected neutrons. The sensitive energy range for the
individual detectors is different though (Köhli et al., 2018). The stochastic uncertainty of the observed neutron count rate $N$ at an arbitrary integration interval $\Delta t$ amounts to $\sqrt{N}/\sqrt{\Delta t}$ (Francke et al., 2022). The uncertainty of soil moisture estimates based on CRNS, however, is subject to a wide range of effects, including the correction for effects of the atmosphere and other hydrogen pools, the collection and weighting of calibration measurements, and the propagation of errors through the non-linear conversion from neutron intensities to soil moisture (see e.g. Jakobi et al., 2020; Weimar et al., 2020; Baroni et al.,
2018; Iwema et al., 2021, for a discussion).

**Table 2.** Properties of CRNS sensors used in the MqC (including manufacturer, model, and detector technology), the availability of detector tubes for epithermal neutrons (moderated - "mod") and thermal neutrons ("bare"), the operation period, the ratio of the sensor's raw counts of epithermal neutrons to the counts of a calibrator sensor (consistent with Fersch et al. (2020); Heistermann et al. (2022a)), referred to as sensitivity.

| ID | Manufacturer | Sensor model | Technology | Tubes | Operated (from - to) | Sensitivity |
|----|-------------|-------------|-----------|-------|---------------------|-------------|
| **1** | Hydroinnova | CRS 2000-B | $^{10}BF_3$ gas | mod+bare | Aug 2019 - Nov 2022 | 1.190 |
| **2** | Hydroinnova | CRS 1000 | $^3$He gas | mod | Aug 2019 - Nov 2022 | 0.452 |
| **4** | Lab-C | HydroSense dual | $^6$Li foil | mod | Aug 2019 - Nov 2022 | 4.544 |
| **11** | Quaesta Instr. | dual BF3-C-4 | $^{10}BF_3$ gas | mod | Nov 2020 - Nov 2022 | 4.754 |
| **21** | Hydroinnova | CRS 2000-B | $^{10}BF_3$ gas | mod | Aug 2019 - Nov 2022 | 1.132 |
| **22** | Hydroinnova | CRS 2000-B | $^{10}BF_3$ gas | mod | Aug 2019 - Nov 2022 | 1.161 |
| **26** | Quaesta Instr. | B-E1-4 | $^{10}BF_3$ gas | mod+bare | Aug 2019 - Nov 2022 | 2.040 |
| **27** | Quaesta Instr. | B-E1-4 | $^{10}BF_3$ gas | mod | Aug 2019 - Nov 2022 | 2.024 |
| 9 | StyX Neutronica | StX-140-5-15 | $^{10}$B-lined | mod | May 2022 - Sep 2022 | 2.277 |
| 11a | Finapp | FINAPP3 | scintillator | mod | May 2021 - Apr 2022 | 0.672 |
| 11b | StyX Neutronica | StX-140-5-15 | $^{10}$B-lined | mod | May 2022 - Nov 2022 | 2.236 |
| 12 | StyX Neutronica | StX-140-5-15 | $^{10}$B-lined | mod | May 2022 - Nov 2022 | 1.834 |
| 13 | StyX Neutronica | StX-140-5-15 | $^{10}$B-lined | mod | May 2022 - Sep 2022 | 2.165 |
| 28 | Hydroinnova | CRS 1000 | $^3$He gas | mod | Aug 2019 - Jun 2020 | 0.459 |
| 30 | Quaesta Instr. | B-E1-4 | $^{10}BF_3$ gas | mod | Jul 2020 - Aug 2020 | 2.040 |

**Bold-IDs** mark the permanent core cluster, non-bold IDs were only operated during shorter time periods (see Fig. 3).

### 3.3 Muons as a reference for incoming neutron intensity

Recent studies have indicated that local counts of muon and gamma particles might have the potential to support methods to correct CRNS data for the variability of incoming neutrons (Stevanato et al., 2022; Gianessi et al., 2022). Conventionally, neutron monitor data are used for that purpose (see Sect. 4.2). In order to allow for further benchmarking studies with regard to
correction approaches, a muon and gamma detector (as part of the scintillator-based sensor FINAPP3, see CRNS sensor 11a in Tab. 2) was operated from May 2021 until May 2022. For detector-related technical details, we refer to Gianessi et al. (2022).

### 3.4 Roving CRNS

Roving CRNS snapshots were acquired as part of the irrigation experiment between July and August 2020 (see Sect. 3.15). The UFZ Hydroinnova rover is a moderated CRNS unit (Hydroinnova LLC, Albuquerque, USA) based on $^3$He gas and has
been used by car, by train, by aircraft, by handwagon, or carried by hand in many previous studies (Schrön et al., 2018; Fersch et al., 2020; Schrön et al., 2021; Heistermann et al., 2022a). Here, we placed the detector in a car to survey the whole MqC area, and then used it on a handwagon to map the irrigated plot and nearby fields inaccessible by car. The uncertainty of the





neutron count measurements from roving follows the same counting statistics as outlined for stationary sensors (Sect. 3.2). The same applies to the uncertainty of corresponding soil moisture estimates. For roving, specific uncertainties arise from the

spatial heterogeneity of hydrogen pools or soil properties, the effect of roads (Schrön et al., 2018), and the trade-off between integration time and spatial resolution (Jakobi et al., 2020; Schrön et al., 2021). All these uncertainties depend on the chosen processing methods and should be discussed by users of this data set.

Measurements were conducted on four days, one before the irrigation events (July 13$^{th}$), and three right after the irrigation events (Jul 23$^{th}$, Aug 6$^{th}$, 11$^{th}$, see Tab. 3). The handwagon has been used in stop-and-go mode with typically 5–10 minutes

residence time per point. Raw data in this repository have been cleaned and contain detector-relevant variables, GPS records, as well as meteorological observations from an external mobile weather sensor mounted on the handwagon. In order to visualize the observations (see Sect. 5.2), the data recorded at intervals of 10 seconds were smoothed temporally with a moving average window of 1 minute, and spatially within a 5 meter radius using the distance-weighting function $W_r$ (Schrön et al., 2017). Further corrections and the conversion to soil moisture followed the procedures outlined in Schrön et al. (2018). Since the

sensor on a handwagon is not shielded by car material, we used a slightly larger calibration factor $N_0 = 13447$ cph compared to other studies.

## 3.5 GNSS-Reflectometry

GNSS-reflectometry (GNSS-R) is a non-conventional methodology, which uses measurements of reflected signals from navigational satellites to estimate soil moisture. The theoretical background was introduced by Larson et al. (2008b, 2009); Rodriguez-

Alvarez et al. (2009b); Zavorotny et al. (2009); Chew et al. (2014), and the efficiency of this method was demonstrated by various research studies (e.g., Larson et al., 2008a; Rodriguez-Alvarez et al., 2009a; Vey et al., 2016).

The GNSS antenna receives the direct signal as well as the signal reflected by the earth's surface. For the purpose of soil moisture retrieval, Larson et al. (2008b) suggested to use the signal-to-noise ratio (SNR) recorded by the antenna, as it is independent of the effects of orbits, atmosphere, or clocks. Eq. 1 in Larson et al. (2008b) provides the key relationship between

the SNR and a phase offset $\phi$. This offset directly relates to the apparent reflection depth of the GPS signal which, in turn, depends on permittivity and hence soil moisture. Accordingly, *relative* soil moisture changes can be retrieved by comparing different phase offsets $\phi$, assuming that other surface properties remain constant. Please note that some more filtering is required to take into account the dominance of the direct signal in high elevation angles (see Larson et al., 2008a, for details).

Actual SWC can then be estimated by relating these relative changes to representative SWC measurements on the ground.

In the MqC context, the obvious choice for such measurements would be the four collocated TDR profiles at a depth of 9 cm (see Sect. 3.8).

Among all constellations, the Global Positioning System (GPS) is the most suitable for GNSS-R soil moisture applications. The repetition period of the orbits is one sidereal day (23 h 56 min 4.0905 s), and each satellite yields two individual tracks over each location. In combination with the track splitting into two arcs (ascending and descending), all GPS satellites provide more

than 100 reflection paths per day, each of them potentially providing a soil moisture estimate. The footprint of the reflection corresponds to a projected ellipse on the ground and is not constant in time. Its shape and size depend on the antenna height,





the wavelength of the carrier frequency, and the elevation angle of the satellite. Its position is related to the unique orbit of the satellite and the antenna height (Larson et al., 2009). For a 3 metre high antenna, the L2 GPS signals yield an ellipse of 33.66 m by 2.93 m size, when the satellite is located 5° above the horizon. The present setup hence accumulates observations from a

circular area of approximately 33 m radius around the station, recorded with a Delta TRE-G3T receiver (JAVAD GNSS, Inc., San Jose, USA) and a S67 antenna (Antcom Corporation, Torrance, USA).

### 3.6 UAS-based hyperspectral remote sensing

In order to be able to observe high resolution SWC patterns and to analyze plant-soil interactions, hyperspectral imagery was acquired during the irrigation experiment (section 3.15). Table 3 displays the four dates of the image acquisitions in the

context of the irrigation experiment. We performed the flights at solar noon (± 1 h) during cloudless sky conditions using a flight altitude of 100 m and a flight speed of approximately 5 m/s. The carrier platform was a DJI Matrice 600 Pro (Da-Jiang Innovations Science and Technology Co., Ltd., China) which is coupled to the Nano-Hyperspec sensor (Headwall Photonics, Inc., USA). This hyperspectral sensor is a linescanner which captures 271 bands in the VIS-NIR spectral range (399 to 1000 nm) with a FWHM of approximately 6 nm. The lense has an angular field of view of 15.3°. Together with the flight altitude this

results in a spatial resolution of approximately 4 cm×4 cm.

We performed the following steps to process the hyperspectral data:

– Conversion of the raw data to radiance and reflectance using the Headwall SpectralView software and a reflectance tarp, which was scanned during the image acquisition.

– Geo-referencing of the reflectance data using the Georeferencer tool of ArcGIS (Esri Inc., USA) and high resolution

multi-spectral UAS data as a basemap. Details of the basemap creation can be found in Döpper et al. (2022).

– Correction of the spectral signal for spikes and drops.

– Spectral filtering of the data with the Savitzky-Golay filter (Savitzky and Golay, 1964) as implemented in the Python SciPy module (Virtanen et al., 2020). We applied a window width of seven and a second order polynomial smoothing.

Besides the processed hyperspectral imagery, we provide the spatial SWC products based on a data-driven approach and a

hybrid approach as described in Döpper et al. (2022). For interpretation and accuracy of the products, we refer to Döpper et al. (2022).

### 3.7 Leaf area index measurements

Over vegetated areas, the hyperspectral signal is dominated by variations of the leaf area index (LAI). We provide LAI measurements as a resource for interpreting the hyperspectral data and disentangling soil moisture-related signals from LAI signals.

The LAI was sampled on the four dates of the UAS-based hyperspectral image acquisition (see Tab. 3) and comprise 31 to 40 measurements per campaign. We measured the LAI using a LAI-2200C Plant Canopy Analyzer (LI-COR Biosciences GmbH, Germany). In order to avoid direct sunlight scattering, we sampled the LAI at dawn, starting shortly after sunset. We randomly





located the measurements to reduce damages of the vegetation due to repetitive sampling. Each measurement resulted from the mean of 5 above and below canopy measurements at each location. The location was recorded using a Leica Zeno GG04
(Leica Geosystems AG, Switzerland) DGPS antenna with accuracies at centimeter level.

## 3.8 Soil moisture profiles

A variety of soil moisture profile measurements was implemented on the premise, covering different measurement depths and technologies (with a total number of 27 individual profiles, 23 of them with more than two years of data). That way, we could obtain detailed records on the vertical SWC dynamics related to infiltration and drying, which are critical to the retrieval and
interpretation of CRNS-derived soil moisture (see e.g. Scheiffele et al., 2020).

Soil moisture profile probes were operated at 12 locations at or close to the CRNS sensors to supplement the CRNS cluster with information on the horizontal and vertical variability of soil moisture (Fig. 2). The impedance-based profile probes (3 PR2/4 and 8 PR2/6, Delta-T Devices LLC, Cambridge, UK) measured at 10, 20, 30, and 40 cm depths (PR2/6 also at 60 and 100 cm). Additionally, 8 profiles consisting of 4–5 single impedance-based probes (ThetaProbe ML2x, Delta-T Devices LLC,
Cambridge, UK) installed at depths down to 200 cm complemented the network in locations where profile probes could not be installed or where larger depths needed to be monitored. According to the manufacturer, using the default conversion function, the uncertainty of the soil moisture measurements amounts to $\pm 5 \, \text{m}^3/\text{m}^3$ (Delta-T, 2016). To improve this conversion, we performed a two-point calibration (air, water) to correct the raw sensor readings and employed a customized function for converting permittivity to volumetric soil moisture, as described in section 3.11. Please also refer to Jackisch et al. (2020) for
a broader assessment of various measurement techniques for soil water content, including impedance-based sensors and the need for recalibration.

Furthermore, 5 profiles of 20 single TDR soil moisture probes (TDR100, Campbell Scientific Ltd., UK) in close vicinity to the permanent GNSS-R antenna are available (Fig. 2). The probes are installed in 9, 11, 25, 45 and 75 cm depth, and operate using the conversion suggested by the manufacturer, i.e., the approach after Topp et al. (1980), to derive volumetric soil water
content in $\text{m}^3/\text{m}^3$ in 15-minute intervals.

Finally, three profiles with measurements in 5, 10, 20, 40 and 60 cm depth were equipped with 5TE sensors (5TE, Decagon Devices, Inc., Pullman, USA) and single ML2x ThetaProbes in the same depth. The single ThetaProbes were calibrated with the above approach and removed in 2020. During installation of the 5TE sensors, soil samples were taken to determine volumetric soil moisture (section 3.11) and derive a linear relationship between sensor permittivity and soil moisture. The sensors also
provide soil temperature measurements; electrical conductivity values were regarded as unreliable and are not provided here.

## 3.9 Groundwater level

Two groundwater observation wells were installed in August 2019 within the MqC and equipped with sensors to continuously record groundwater heads. The depth to the groundwater table increases from very shallow (open water body of the Wublitz, 200 m west of MqC) towards approximately 10 m below ground in the east of MqC. The wells are located close to CRNS
sensor 22, approximately at the middle of the hillslope (ground elevation 35.9 m a.s.l.), and in the vicinity of CRNS sensor 2





at the bottom of the hillslope (31.2 m a.s.l.) (Fig. 2). Well pipes of 63 mm diameter were installed at a depth of 4 m and 6 m below ground for well 2 and 22, respectively. The pipes are filtered over a length of one meter at the lower end. A pressure sensor (Hobo U20L, Onset) was installed in the well 2 and well 22 at 3.6 m and 5.8 m below ground, respectively. Both sensors recorded pressure and temperature at a 30 min interval. In well 22, an additional sensor recorded air pressure and temperature.

Regular manual measurements of the groundwater heads were used to validate the continuous measurements and exclude any drift in the pressure sensor measurements.

### 3.10 Measurement of matric potential (tensiometer)

At locations 2, 11, and 22 (Fig. 2), tensiometer profiles were installed in four depths in June 2020. Alongside each tensiometer, a ThetaProbe (ML2x, Delta-T Devices LLC, Cambridge, UK) measured the soil water content (see section 3.8). Installation

depths at location 11 and 22 were 50, 100, 150, 200 cm. At location 2 the depths of 50, 80, 110 and 140 cm were chosen because of shallow groundwater conditions. The matric potential was measured with full range tensiometers (TensioMark, echoTech Umwelt-Meßsysteme GmbH, Bonn, Germany) which indirectly measure the matric potential via heat pulse dissipation. During installation, three soil cores were taken in each depth. One core ($100 \, \text{cm}^3$) served to determine water content and texture (see section 3.11). Two larger cores ($250 \, \text{cm}^3$) were used to measure soil hydraulic properties in the laboratory (see section 3.12).

### 250 3.11 Campaign-based observation of soil water content and other soil data

In order to increase the spatial density of independent soil moisture observations in addition to the stationary profile probes, two sampling campaigns were carried out on November 7, 2019 and October 10, 2022. These campaigns involved a total of 103 (in 2019) / 87 (in 2022) locations with manual measurements using soil cores (35/13 sampled locations) and ThetaProbes probes (72/87 sampled locations), typically down to a depth of 30 cm with increments of 5 cm. All sampling locations were

surveyed by dGPS (see Fig. 2 for an overview). In October 2020, we retrieved another 29 soil cores specifically for the analysis of bulk density, organic matter, lattice water content and texture. Additionally, during the installation of the tensiometers (see Sect. 3.10) and the 5TE sensors (Sect. 3.8), soil cores were collected from the installation depths down to to 2 m, allowing the retrieval of the mentioned parameters for greater depths.

Soil cores were extracted and treated as described in Heistermann et al. (2022a). Water content and bulk density were derived

by oven drying, organic matter and lattice water determined on subsamples by loss-on-ignition. The texture analysis was done on untreated soil samples by wet-sieving (for gravel, sand) and laser diffraction (silt, clay). The manual measurements with ThetaProbes, too, followed the procedure outlined by Heistermann et al. (2022a), using portable ThetaProbes with sensor-specific calibrations and a site-specific conversion of permittivity to soil moisture, which resulted into an RMSE of $0.03 \, \text{m}^3/\text{m}^3$ for the soil moisture estimates.

In addition to the two campaigns in November 7, 2019 and October 10, 2022, extensive near-surface measurements (ThetaProbes) were conducted on 13 days between July 18, 2019, and August 17, 2020. These comprised fivefold replicates at 35 to 71 random locations with electrodes inserted from the surface (i.e., effective measurement depth 2.5 cm), processed as described above.



## 3.12   Evaporation experiment to determine soil hydraulic parameters

Two undisturbed soil cores of $250\,cm^3$ were taken during the installation of the three tensiometer and deep soil moisture profiles at each installation depth (see section 3.10). The resulting 24 soil cores underwent soil hydraulic analyses in the laboratory. First, saturated hydraulic conductivity was measured by the constant head method (Klute and Dirksen, 1986). Second, the cores were used to determine the unsaturated soil hydraulic properties within an evaporation experiment (Wind, 1968; Schindler and Müller, 2006). This was done using a kupF MP10 device (UGT, Umwelt-Geräte-Technik GmbH, Müncheberg, Germany). At

an interval of ten minutes the device records the weight of the soil core as well as the matric potential at two tensiometers vertically inserted at depths of 1.25 and 3.75 cm. Afterwards, dry weight and bulk density of the samples were determined. Additionally, the water content at a pF-value of 4.2 (or suction of 15,000 hPa) was determined using a ceramic plate and pressure chamber on two small subsamples per soil core (Brooks and Corey, 1964; Klute, 1986). The data can be used to determine the retention curve and unsaturated hydraulic properties of the soil (e.g. Peters and Durner, 2008).

## 3.13   Measurement of snow depth, density and cover

In MqC, an appreciable snow cover can be expected every two to three years. In February 2021, a snow layer with a maximum depth of approx. 10 cm persisted for about one week. During this time, its main properties were measured: snow depth (continuously and campaign based), precipitation, snow cover and density (campaign-based).

Snow depth was monitored with an ultrasonic temperature-compensated distance-meter (SR50-45, CampbellScientific, Inc.,

Logan, USA) looking downward on a representative spot of shortly-clipped grass. Simultaneously, close to three CRNS sensors (1, 2 and 11, see Fig. 2) wildlife cameras (SECACAM HomeVista, VenTrade GmbH, Köln, Germany) with infrared night vision took hourly images of sets of ten 50-cm snow stakes, distributed in clusters of approx. 10 m diameter each. The length of the stakes protruding from the snow was determined manually on the resulting images and used for a simple photogrammetric calculation of time series of the snow depths at the stakes.

Shortly before and during the snow period, total precipitation and air temperature was additionally measured using a weighting-based pluviometer Pluvio (Pluvio2 L, Ott GmbH, Kempten, Germany and 107-L Temperature Probe, Campbell Scientific, Inc.) with anti-freezing agent to reduce losses by spindrift.

Areal snow cover during snow-melt was surveyed on two dates (February 17 and 18, 2021) by UAS imagery using a Mavic Pro (Da-Jiang Innovations Science and Technology Co., Ltd, China) at a flight altitude of 100 m. Mosaicking employed

Photoscan software (Agisoft LLC, St. Petersburg, Russia) and was geo-referenced to 20 cm orthophotos provided by the Federal State of Brandenburg, yielding an RGB image with 3 cm ground resolution.

Manual snow sampling included snow depth measurements with a ruler and density measurements. These were conducted using cylinder cores or collecting (sweeping up, rolling) all snow from a designated area and successive weighing in the field.



## 3.14 Land use and biomass

Hydrogen as contained in biomass affects both epithermal and thermal neutron count rates. The corresponding biomass pools in the sensor footprints should hence be accounted for when interpreting the CRNS signal.

For the agricultural plots, the crops, date of operation and their yields have been determined by weighing the harvest. From these yields, total biomass inventories were computed using literature values for water content and harvest index (Munns et al., 2018; Stöckle et al., 2022; Taes et al., 2022; Kuai et al., 2015).

The highest biomass densities in and around the study area are within the surrounding forests in the west and north (Fig. 2), followed by the orchards (cherry and apple plantations) and a short rotation plantation of poplar trees. For these areas of higher biomass density, the determination of the average above-ground dry biomass (AGB) was further refined by using allometric functions. In the forest stands and the poplar plantation, we used *in situ* measurements of diameter at breast height (DBH, in cm) and tree height (H, in m) and species-specific allometric relationships to predict AGB (in kg) per tree. To select an appropriate allometric relationship and its parameter values, we chose published studies with similar climate and a similar range of reported DBH values. The AGB was summed up for all trees in a sampled plot and divided by the plot area to obtain the AGB density (in kg/m$^2$).

- Forest: The adjacent northern forest area is predominantly covered by black locust (*Robinia pseudoacacia*, 79 % of sampled trees), oaks (*Quercus robur* L.; *Quercus petraea* (Matt.) Liebl.) and maples (*Acer pseudoplatanus*, *Acer platanoides* and *Acer campestre*). The western forest stand, part of a local nature reserve, is dominated by alder (*Alnus glutinosa*, 89 %) with scattered presence of elder (*Sambucus nigra*) and elm (*Ulmus* spp.). Allometric relationships were taken from Carl et al. (2017) (for *Robinia pseudoacacia*), Wang (2006) (*Acer*), Zianis et al. (2005) (*Alnus* and *Quercus*), Kort et al. (2011) (*Sambucus nigra*) and Clark et al. (1986) (*Ulmus sp.*). Within each forest area intersecting with a CRNS footprint, we randomly selected 3 plots with a radius of 12.5 m and measured DBH (at 1.3 m reference height) and H (using a laser-based ranging device, TruPulse 360B, Laser Technology, Inc., Centennial, USA). Herbaceous plants, litter and understory vegetation were not taken into account as these represent a minor hydrogen pool as compared to the adult trees. An average of 6.1±4.0 kg/m$^2$ was estimated for the northern forest and 9.0±3.0 kg/m$^2$ in the western forest, respectively.

- Short rotation poplars: The poplar plot (with an area of 3227 m$^2$) consists of a total of 12 rows, equally divided between 3 clones (MAX 4, NE42 and Matrix 24). Spacing between the trees was 0.5 m and there were approximately 720 trees in the plot. A total of 60 trees in 6 adjacent rows were sampled for DBH and H. The equation for the allometric relationship for *Populus* was taken from Zell (2008). On average, the biomass in the poplar stand was estimated to be 1.3±0.6 kg/m$^2$.

- For the cherry and apple orchards, we refer to Richter (2021) who measured the average AGB per tree from a representative sample of single trees. Combined with the total number of trees per plot and the plot size, the average AGB density yields to 3.8±0.4 kg/m$^2$ for the cherries and 0.49±0.08 kg/m$^2$ for the apples.





**Table 3.** Data acquisition during irrigation experiment, July–August 2020

| | July | | | | | | | | | | | | | | | | | | | | August | | | | | | | | | | | | | | | | | |
|---|---|---|---|---|---|---|---|---|---|---|---|---|---|---|---|---|---|---|---|---|---|---|---|---|---|---|---|---|---|---|---|---|---|---|---|---|---|---|
| | 13 | 14 | 15 | 15 | 16 | 17 | 18 | 19 | 20 | 21 | 22 | 23 | 24 | 25 | 26 | 27 | 28 | 29 | 30 | 31 | 01 | 02 | 03 | 04 | 05 | 06 | 07 | 08 | 09 | 10 | 11 | 12 | 13 | 14 | 15 | 16 | 17 | 18 |
| irrigation | | | | | | | | | | | ✔ | ✔ | | | | | | | | | | | | | ✔ | ✔ | | | | | ✔ | | | | | | | |
| stationary | ✔ | ✔ | ✔ | ✔ | ✔ | ✔ | ✔ | ✔ | ✔ | ✔ | ✔ | ✔ | ✔ | ✔ | ✔ | ✔ | ✔ | ✔ | ✔ | ✔ | ✔ | ✔ | ✔ | ✔ | ✔ | ✔ | ✔ | ✔ | ✔ | ✔ | ✔ | ✔ | ✔ | ✔ | ✔ | ✔ | ✔ | ✔ |
| rover | ✔ | | | | | | | | | | | ✔ | | | | | | | | | | | | | ✔ | | | | | | ✔ | | | | | | | |
| gamma | | | | | | | | | | | | ✔ | | | | | | | | | | | | | ✔ | | | | | | ✔ | | | | | | | |
| profiles (perm.) | ✔ | ✔ | ✔ | ✔ | ✔ | ✔ | ✔ | ✔ | ✔ | ✔ | ✔ | ✔ | ✔ | ✔ | ✔ | ✔ | ✔ | ✔ | ✔ | ✔ | ✔ | ✔ | ✔ | ✔ | ✔ | ✔ | ✔ | ✔ | ✔ | ✔ | ✔ | ✔ | ✔ | ✔ | ✔ | ✔ | ✔ | ✔ |
| profiles (temp.) | | ✔ | | | | | | | | | ✔ | ✔ | | ✔ | | | | | | | | | | | ✔ | ✔ | | | | ✔ | ✔ | | | ✔ | | | | |
| surface | | ✔ | | | | | | | | | ✔ | ✔ | | ✔ | | | | | | | | | | | ✔ | ✔ | | | | ✔ | ✔ | | | ✔ | | | | |
| hyperspectral | | | | | | | | | | | | | | | | | | | | | | | | | ✔ | ✔ | | | | | ✔ | | | ✔ | | | | |
| LAI | | | | | | | | | | | | | | | | | | | | | | | | | ✔ | ✔ | | | | | ✔ | | | ✔ | | | | |

– Other plots: The walnut plot and a row of cornel cherries are the only appreciable woody vegetation for which no biomass or allometric data were collected as part of this study. Nevertheless, for the purpose of correcting the CRNS signal, we used the biomass estimate of the cherry plot as an acceptable approximation.

### 3.15 Irrigation Experiment (July–August 2020)

An irrigation experiment was carried out in order to observe the response of various sensors to pronounced soil moisture contrasts. Between July 22 and August 11, a plot of 35 m × 100 m (Fig. 2) was repeatedly irrigated with a centre pivot sprinkler (Irriland, Italy): During three phases, 40–50 mm were applied to 3/3, 2/3 and 1/3 of the plot, respectively, i.e. the southernmost part receiving three pulses of water (see Tab. 3). Three rain gauges and one pan (41 cm × 72 cm) served for verifying the respective amount of applied water. Accompanying measurements included the following features (see also Tab. 3):

– Three CRNS sensors were operated at close proximity to the irrigated plot, including sensors 1 and 22. The CRNS sensor from location 26 was temporarily relocated to the eastern edge of the plot (location 30) since the CRNS sensors at locations 11 and 13 were not operated at the time.

– Nine campaigns of manually measured soil moisture at the surface (at 46–71 locations, see "near surface measurements" in section 3.11), and in 20 PR2-access tubes. The access tubes were visited nine times with a mobile PR2-profile-probe,
taking three readings at 0, 120 and 240 degree orientation. Three of the 20 access tubes were also continuously monitored throughout the experiment.

– Four mobile CRNS roving campaigns (see section 3.4 for details).

– Four UAS flights to acquire hyperspectral imagery (see section 3.6) for potential mapping of SWC patterns within the irrigated field.

– Four leaf area index measurement campaigns (see section 3.7) to complement the remote sensing observations since LAI is one of the most dominant variables in the spectral signal.



## 4 Relevant data provided by third parties

The following subsections highlight relevant data sets which have been published already or are provided by other organisations or channels, but which we consider as potentially helpful for users of the data presented in section 3.

### 355 4.1 Weather data

A climate station with an heated tipping bucket rain gauge is located in the north-eastern part of the study area (see Fig. 2) and recorded, at an hourly resolution, standard climate variables, including air temperature, relative humidity, precipitation, soil temperature (at 5, 10 and 30 cm), solar irradiation, as well as wind speed and direction. The original data is openly available at http://technologygarden.atb-potsdam.de/bsa_wetter.aspx.

The closest climate station of the German Weather Service (Deutscher Wetterdienst, DWD) is located in Potsdam about 12 km south-east of the study site (station ID 03987). The data is available via DWD's open data repository https://opendata.dwd.de/climate_environment/CDC/observations_germany/climate/.

For convenience, we included subsets of both station records in the published dataset, spanning the MqC study period.

### 4.2 Incoming neutron flux

Variations of the incoming cosmic-ray neutron flux on Earth are recorded by neutron monitors. The corresponding data are available from the Neutron Monitor Database, http://www.nmdb.eu. Following Hawdon et al. (2014); Schrön et al. (2016); Baatz et al. (2015); Jakobi et al. (2018); Baroni et al. (2018), the neutron monitor at Jungfraujoch (JUNG) is recommended as for the correction of the incoming neutron flux at MqC.

### 4.3 Terrain and soil maps

A digital elevation model (DEM) at the resolution of $1\,\text{m} \times 1\,\text{m}$ is freely available at the Landesvermessung und Geobasisinformation Brandenburg (LGB, state survey agency) at https://geobroker.geobasis-bb.de, at an accuracy of 30 cm. A soil map for agricultural soils (Mittelmaßstäbige landwirtschaftliche Standortkartierung) is openly available at a scale of 1:100,000 at https://geoportal.brandenburg.de.

### 4.4 Land use, roads, waterways

During fieldwork and for visualisation, we used OpenStreetMap data layers (OpenStreetMap contributors, 2023) available via http://download.geofabrik.de, namely landuse, waterways, and traffic ways. The data are distributed under ODbL license (www.openstreetmap.org/copyright).


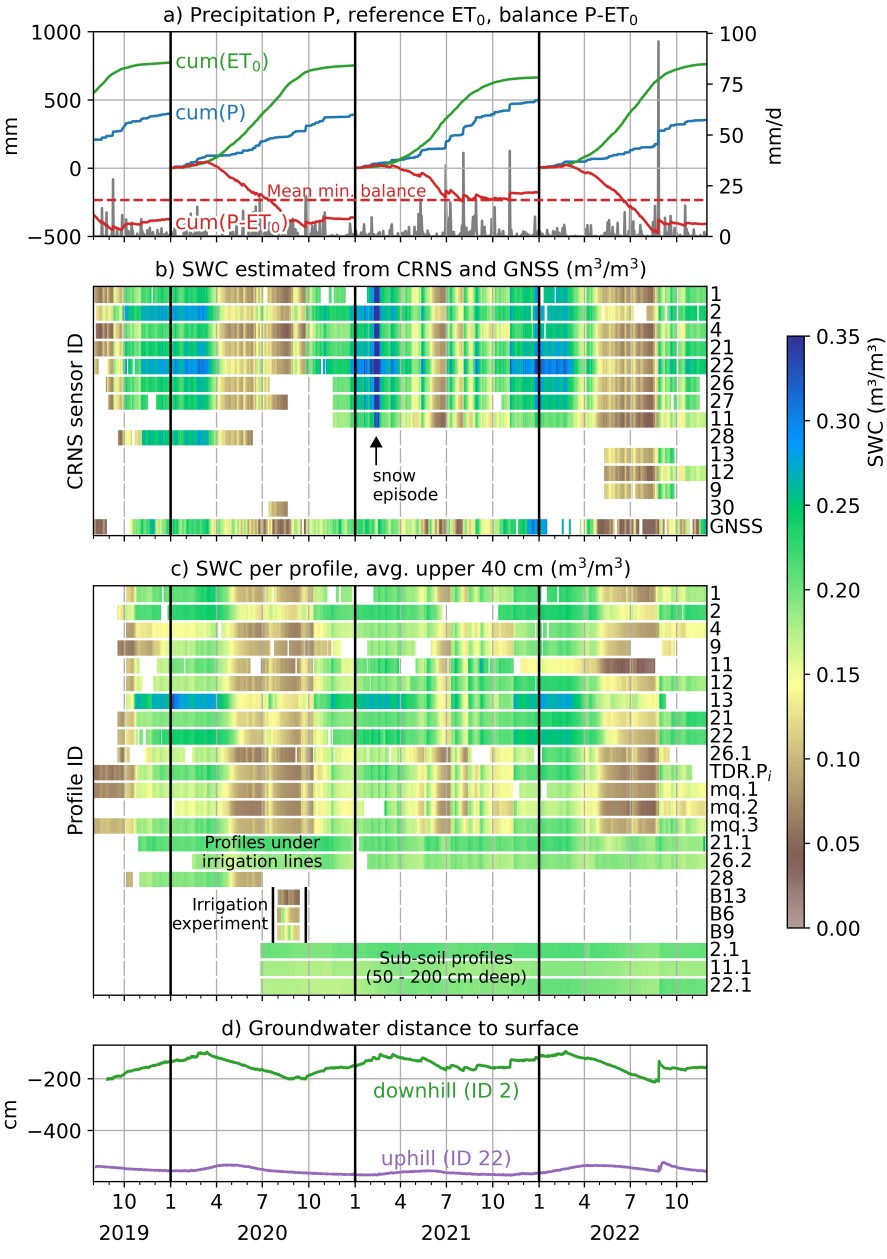

**Figure 3.** Temporal dynamics of recorded variables and data availability from August 2019 to November 2022; a) daily precipitation (grey bars), cumulative sums of precipitation P, reference evapotranspiration $ET_0$, and the daily difference P-$ET_0$, separately for each calendar year; b) soil moisture estimated from neutron count rates (CRNS) and from the GNSS signal (temporary CRNS sensors 11a and 11b (Tab. 2) not shown as we focus on an overview of locations, not sensors; c) SWC as recorded at the profiles, averaged over upper 40 cm; label TDR.$P_i$ refers to the average of five TDR profiles closely collocated at the GNSS antenna; d) groundwater distance below ground at two wells.





## 5 Exemplary views at the data

A comprehensive presentation and discussion of this large and diverse data set is beyond the scope of this paper. Still, this
section provides an overview, and also highlights selected details. This involves some level of data processing, specifically
with regard to the correction of neutron counts from stationary sensors and CRNS roving, as well as the calibration of the
relationship between neutron counts rates and volumetric soil moisture. Details with regard to the corresponding processing
steps are available e.g. in Heistermann et al. (2021, 2022a).

### 5.1 An overview of the entire study period

Figure 3 provides a summary of the time series recorded from 2019 to 2022. It illustrates both temporal dynamics and data
availability for the various locations and sensors.

The study period coincided with three major drought years (2019, 2020 and 2022). This becomes apparent from the cumulative difference of precipitation and reference evapotranspiration (Fig. 3a) as well as the soil moisture obtained from the
CRNS (Fig. 3b) and the soil moisture profiles (Fig. 3c). While the 2019 drought period was only captured in late summer with
the launch of MqC, the droughts in 2020 and 2022 were covered in their full extent, starting already in May and ending in
September.

The dynamics of the groundwater level data (Fig. 3d) are consistent with both the cumulative sum of P-ET$_0$ and the soil moisture estimates. Specifically, the groundwater level at the hill foot (ID 2) shows distinct seasonal dynamics, with groundwater
recharge typically starting in September. In August 2022, an exceptional heavy rainfall event caused an immediate groundwater
level response, which is discussed in section 5.3 in further detail.

In February 2021, the CRNS-based soil moisture estimates (Fig. 3b) exhibited exceptionally high values. This was caused by
a period of substantial snow cover, the only such episode in the study period. The effects of snow have not yet been corrected
for in the CRNS-based soil moisture estimation. However, independent snow measurements (see section 3.13) in space and
time are available to explore options for estimating both soil moisture and snow water equivalent from CRNS data, e.g. by
exploiting thermal neutron counts.

### 5.2 Irrigation experiment

Fig. 4 highlights the irrigation experiment that took place in July and August 2020. The upper four panels contrast snapshots
of spatial soil moisture estimates from CRNS roving, UAS-based hyperspectral remote sensing, profile probes and surface
measurements. The different data density in space and time is apparent, however, in-depth intercomparisons of the different
approaches could lead to more insights into their strengths and weaknesses.

Fig. 4q-r show the continuous records obtained from the closest three CRNS probes and one selected profile-probe in the
irrigated field (ID B9). The profile probe shows a clear response to the first and second irrigation pulse. As the third pulse
only covered the southernmost third of the test area, it is not reflected in the profile probe. The irrigated volume of 40–50 mm



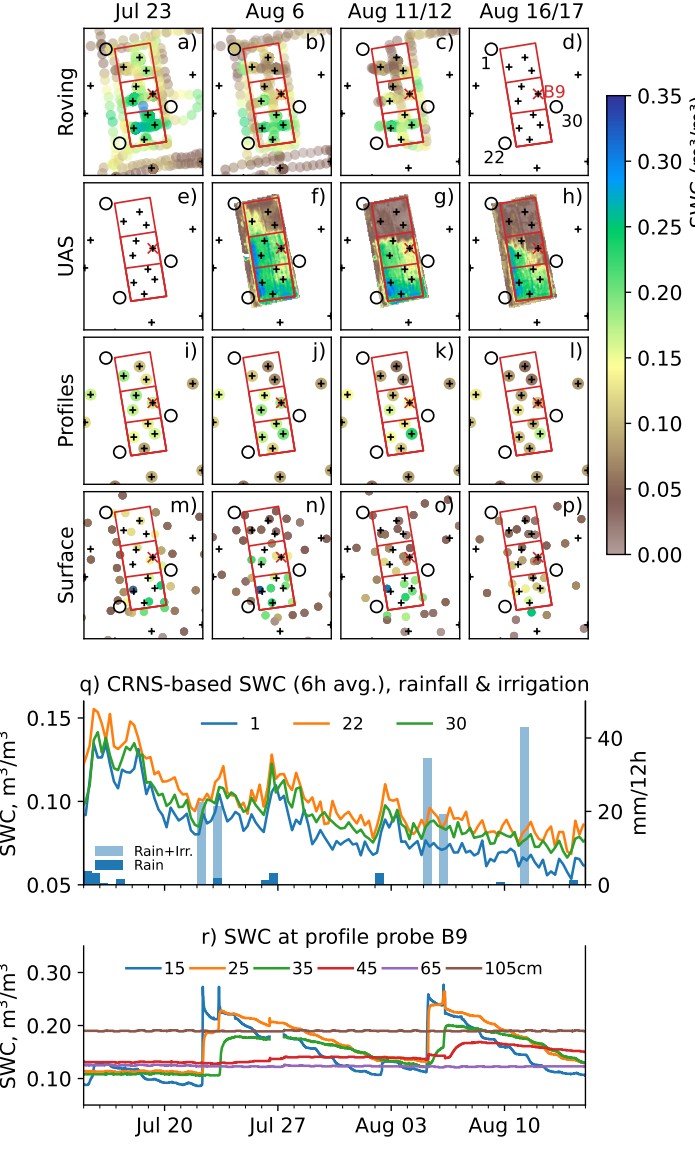

**Figure 4.** Soil moisture observations and products during the irrigation experiment in July and August 2020; a-d) CRNS roving; e-h) hyperspectral UAS-imagery; i-l) intermittent PR2 (impedance-based) profile measurements, averaged over the upper 40 cm; m-p) intermittent ThetaProbe measurements at the soil surface; q) stationary CRNS sensors, based on neutron counts average over six hour intervals locations marked by black circles; r) stationary profile probe B9, location marked by red cross.

affects soils moisture in the upper 30–40 cm, while it remains constant below 60 cm. The effects of the smaller rainfall events
are merely captured by the uppermost sensor (15 cm).





In contrast, even small rainfall events caused remarkable rise in the CRNS-based soil moisture, demonstrating its pronounced sensitivity to changes in shallow soil depths. The neutron response to the irrigation, in turn, remains relatively small: even for the first irrigation event, the irrigated plot only constitutes a small fraction of the entire CRNS footprint. This becomes even more evident for the second and third irrigation events. Prospective research will show whether it is, despite this low signal to
noise ratio, possible to resolve the subfootprint heterogeneity introduced by irrigation (see also Brogi et al., 2022).

### 5.3 Heavy rainfall

The observation period comprised six heavy rainfall events with daily totals over 30 mm (Fig. 5a-f), including an extreme event with almost 100 mm per day and a maximum hourly depth of 60 mm in August 2022 (Fig. 5f). Not surprisingly, all events clearly show in the signal of the CRNS sensors (Fig. 5g-l). The change in CRNS-based soil moisture not only depends on the
total precipitation depth of the event, but also on the effective event duration: e.g. the short event on August 4, 2021, causes a smaller soil moisture change than the event on June 30, 2021, although the total depth of 53 mm is the same. Interestingly, the soil moisture differences between some CRNS locations can be subject to considerable intra-event dynamics. While sensor 2 tends to be the wettest location before the event, it often ranks closer to the median of the ensemble after the rainfall while sensor 22 shows the opposite behaviour. It remains to be shown whether these changes are in fact caused by location-specific
hydrological processes or whether they could also result from specific parameter constellations along the conversion from neutron count rates to volumetric soil moisture.

Fig. 5m-r shows the soil moisture profiles, averaged per measurement depth from 10 to 100 cm. Below 40 cm, there is usually no remarkable response of soil moisture during or immediately after a rain event. The event on August 26, 2022, however, shows a remarkable signal over all monitored depths. Furthermore, all depths from 10 to 100 cm respond almost immediately,
which suggests fundamentally different processes of the vertical flow (similarly, but less pronounced, on August 4, 2021). The exceptional event on August 26, 2022, even propagates down to the groundwater table for both the uphill and the downhill groundwater level observations (Fig. 5x). Rainfall events of this magnitude could produce surface runoff at MqC, however, only sporadic visual evidence is available, e.g. for the August 26 event.

### 5.4 Snow

Fig. 6 summarizes the data acquired during the snow monitoring phase in 2021. The snow cover formed when temperatures had dropped below zero and precipitation started after February 7. While the weighing-based pluviometer (rain gauge 1 in Fig 6a) registered 5 mm, the permanent precipitation gauge (rain gauge 2) recorded significantly less, probably because of insufficient or malfunctioning heating and resulting of wind drift from the device (Fig. 6a). The resulting snow cover at the snow gauge peaked at around 10 cm (Fig. 6d), while the maxima at the snow stakes varied between 8 and 27 cm. The snow cover caused
a marked decrease in the CRNS count rates (Fig. 6b). In contrast, the prior rainfall event (February 4) which yielded almost three times the precipitation height, resulted in a much lower reduction in the neutron counts. During the approximately 10 days of snow cover, snow depth, soil moisture (Fig. 6c) and neutron counts remained comparatively stable (except CRNS 2). This phase ended with a rapid thawing: the snow melted almost completely within less than two days (except for the western

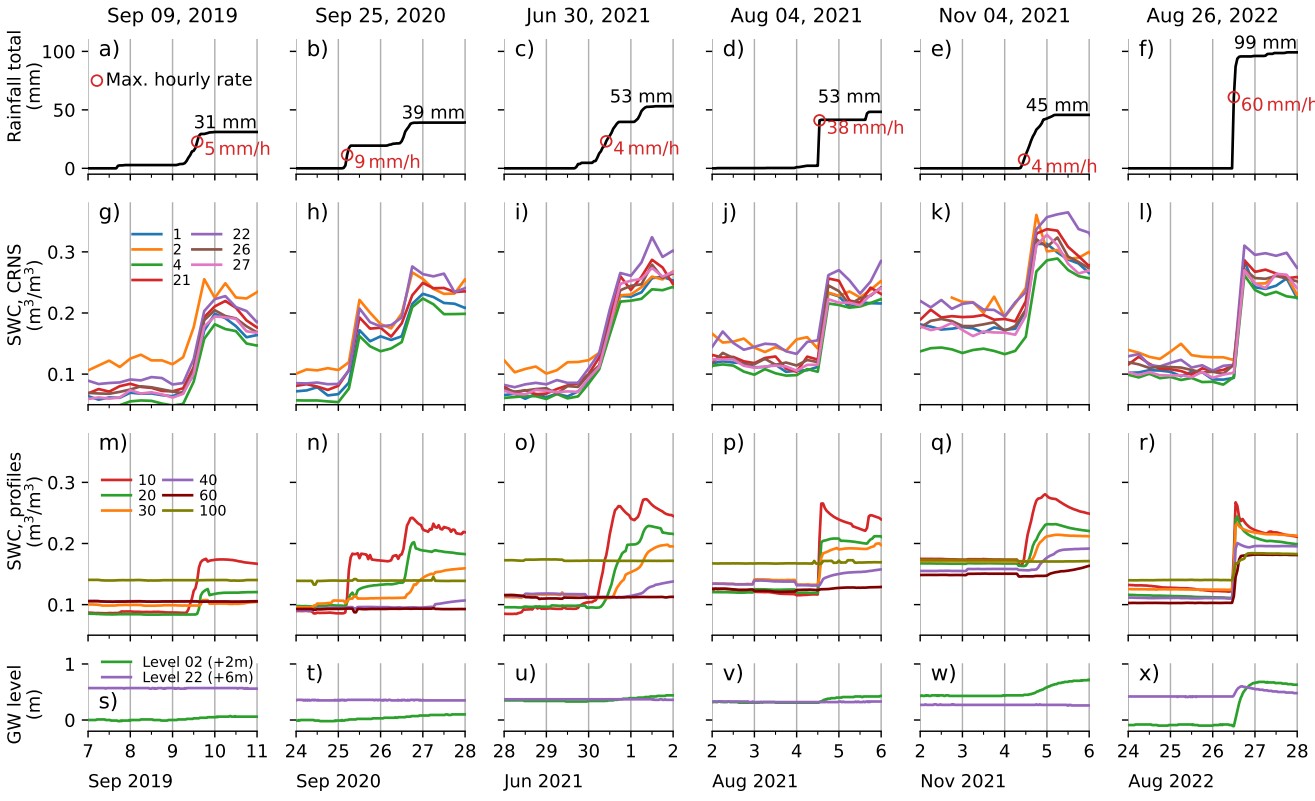

**Figure 5.** The six events with the highest daily rainfall depths during the study period. a-f) cumulative hourly rainfall depth; g-l) soil moisture from CRNS-probes; m-r) soil moisture from profile probes, averaged over all locations; s-x) groundwater (GW) levels.

margins of the MqC and some leeward structures), resulting in another increase in soil moisture by melt water, very similar to the response of the prior rainfall event. The spatial heterogeneity of snow depth in the thawing phase is represented not only by manual measurements, but also by two UAS-borne acquisitions of optical imagery on February 17 and 18 (Fig. 6e and f). Altogether, the comprehensive monitoring of a complete buildup and thawing cycle should provide an excellent research opportunity to investigate the interplay of vertical soil moisture distribution and snow cover on the CRNS signal.

## 6 Conclusions

From August 2019 to November 2022, eight CRNS sensors, 23 permanent and four temporary SWC profiles, two groundwater gauges and one climate station were almost continuously operated in an area of 10 ha at the agricultural research site in Marquardt, Germany. If aggregated, the eight CRNS core sensors - some among the most sensitive ones available for stationary CRNS detectors - provide a neutron count rate about 37 times higher than the one of a conventional Hydroinnova CRS-1000, and hence an unique signal-to-noise ratio for continuous measurements at this spatial scale.

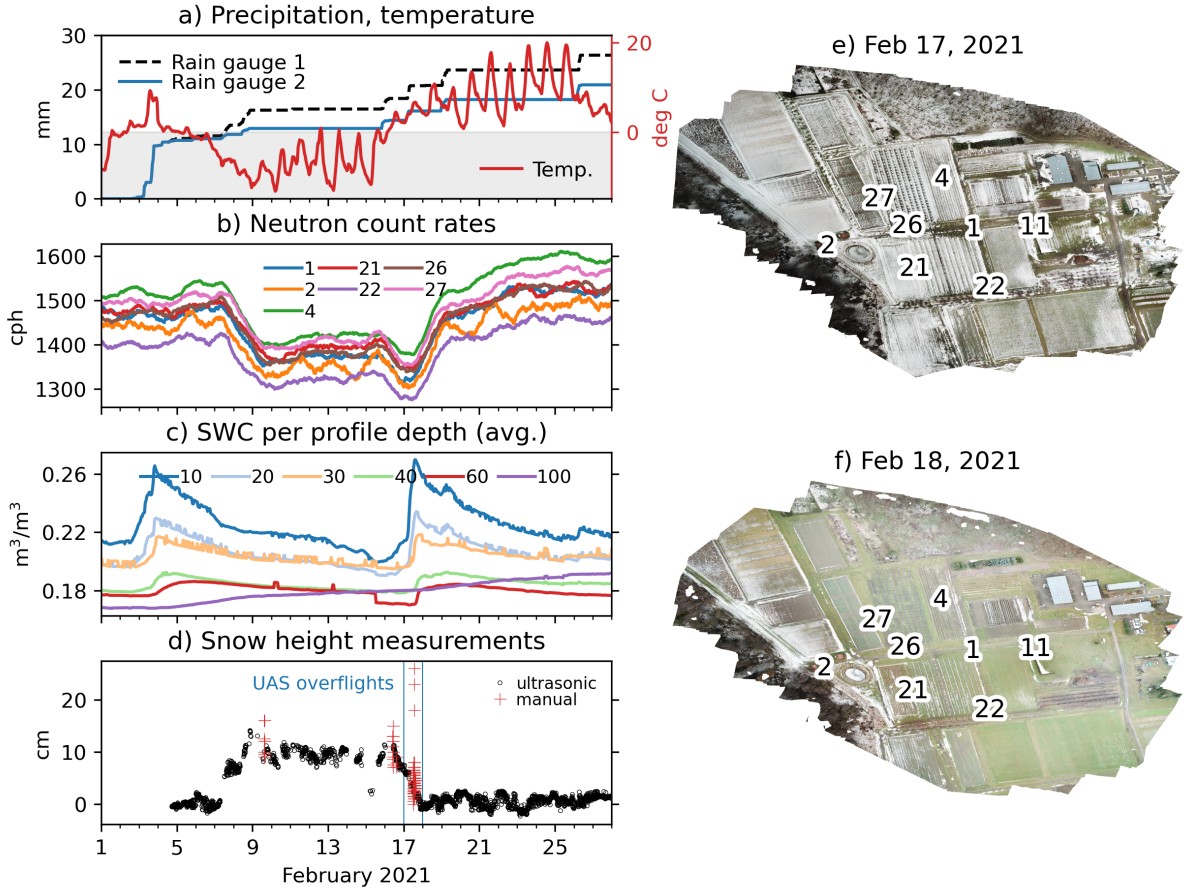

**Figure 6.** Spatial and temporal patterns of snow occurrence in February 2020. a) Time series of air temperature and cumulative precipitation (rain gauge 1: weighing-based pluviometer installed for the snow period; rain gauge 2: permanent pluviometric station installed at Marquardt site); b) corrected (pressure, humidity, incoming) epithermal neutron counts; c) soil moisture from profile probes, averaged over all locations; d) time series of snow depth from ultrasonic snow gauge and manual measurements, e and f) optical UAS-based imagery representing spatial snow heterogeneity during melting.

The core series were supplemented by a wide range of additional measurements:

- Additional CRNS sensors and SWC-profile measurements were implemented for shorter time periods (weeks to months). Some of these additional measurements aimed to cover, at least temporarily, additional locations or soil depths (e.g., during the irrigation experiment). Others were co-located with the core sensors in order to allow for instrument comparisons;

- Various intensive snapshot campaigns, including a large number of manual soil moisture measurements for ground
truthing and soil mapping, CRNS roving and UAS-based hyperspectral remote sensing as part of an irrigation experiment, intensive monitoring and mapping of snow depth, density and coverage, and biomass mapping in areas with an





aboveground biomass density substantially higher than observed for field crops (orchards, poplar plantation, adjacent forests);

– Complementary time series of additional variables and sensor systems, including GNSS-R, tensiometers, and soil temperature sensors;

– Detailed records of land use and irrigation management as part of the agricultural operations.

We extensively documented this data set and exemplarily highlighted various interesting features, including the representation of soil moisture from CRNS and profile probes during highly contrasting conditions such as drought, irrigation, heavy rainfall, and snow coverage. The long monitoring period of three years is a prerequisite to explore the sensor response to and the impact of such contrasting conditions with statistical significance.

This comprehensive data set provides the opportunity to investigate a diverse set of research problems, such as:

– the effects of vertical and horizontal soil moisture on the CRNS signal, in combination with the effects of biomass heterogeneity and snow cover;

– the retrieval of spatial soil moisture patterns from dense CRNS observations under consideration of different governing processes (irrigation, soil variability, cropping patterns) and with different levels of auxiliary data (e.g. CRNS roving, hyperspectral remote sensing, soil water modelling);

– the response of different CRNS sensor types that were directly collocated (e.g., at location 11);

– preferential and bypass flow as well as surface to groundwater connectivity under heavy rainfall conditions;

– the potential of different sensor combinations to produce representative soil moisture estimates for heterogeneous landscapes, which could serve as a reference for remote sensing or hydrological modelling.

Consequently, the application of this dataset is not limited to the CRNS community, it can also serve as a valuable resource to various neighbouring disciplines, including soil and groundwater hydrology, agriculture, remote sensing and hydrological modelling. Currently, MqC is in the process of being extended: in addition to the dense core network of eight CRNS sensors, positions are being re-arranged and up to eight additional sensors are in the process of being added to achieve a coverage of a total area of at least $0.5\,\mathrm{km}^2$. This modification also implies that November 2022 is a natural end point of the McQ data set in the configuration presented here.

## 7 Data availability

The published data set is organized along instruments and observed variables, and follows the structure of Sect. 3 of this paper (Tab. 4). Each subset of data is documented in a dedicated meta-data file in "json" format. Format conventions follow Fersch et al. (2020) and Heistermann et al. (2022a) and are summarized in a 'readme' file. We used EUDAT infrastructure

(https://eudat.eu), namely the services B2SHARE and B2HANDLE, in order to manage identifiers and guarantee long-term persistence. The repository's reference is https://doi.org/10.23728/b2share.edfdaa0d2a82477fa512bde3f53312f2 (Heistermann et al., 2022b).

**Table 4.** Structure of the data repository, and the relation of data subsets to the subsections of this paper.

| Section | Observation | Data subset in the repository |
|---------|-------------|-------------------------------|
| 3.2 | Stationary CRNS | crns_stationary.zip |
| 3.3 | Muon data | crns_muons.zip |
| 3.4 | Roving CRNS | crns_roving.zip |
| 3.5 | GNSS-R | gnss-r.zip |
| 3.6 | Hyperspectral remote sensing | hyperspectral.zip |
| 3.7 | Leaf area index | lai.zip |
| 3.8 | SM-profiles | soilmoisture_profiles.zip |
| 3.9 | Groundwater level | groundwater.zip |
| 3.10 | Tensiometer | soilmoisture_tensiometer.zip |
| 3.11 | Manual soil sampling | soilmoisture_manual_sampling.zip, soilmoisture_surface_sampling.zip |
| 3.12 | Soil hydraulic properties | soil_hydraulics.zip |
| 3.13 | Snow data | snow.zip |
| 3.14 | Landuse / biomass | landuse_biomass.zip |
| 3.15 | Irrigation experiment | irrigation_experiment.zip |

*Author contributions.* TF, MH, LS, and SO designed the study and coordinated the instrumentation; TF and MH coordinated the data management and led the writing of the manuscript; TF processed the soil moisture and snow measurements; MH processed the CRNS data and prepared Figs. 2–6; LS processed the groundwater and soil hydraulic data; KDP processed the tensiometer and quantified the biomass; CB was responsible for setting up and maintaining the instrumentation and supported the data management; MS designed the rover campaign and processed the rover data; BT co-designed the instrument network and provided data on land use, yields and irrigation water use; DR and AG provided the data for the TDR-based soil moisture profiles; VD and MF provided the UAS-based hyperspectral data and related soil moisture products and LAI measurements; MK processed the data from the StyX sensors and supported their maintenance; LA developed the concept and implemented Fig. 1; NA processed the GNSS data; MZS supported the biomass quantification in the cherry and apple plantation; SO headed the MqC effort and is the principal investigator of this study; all authors contributed to writing and proofreading the manuscript.

*Competing interests.* Markus Köhli holds a CEO position at Styx Neutronica GmbH.



*Acknowledgements.* This research was funded by the Deutsche Forschungsgemeinschaft (DFG, German Research Foundation) – research
505 unit FOR 2694 "Cosmic Sense", project number 357874777. We thank Marie-Therese Schmehl for helping in the processing of the snow
cover imagery and a number of student assistants for acquiring field data. We greatfully acknowledge the support from Finapp S.r.l., Padova,
Italy, providing a FINAPP3 CRNS-muon sensor for operation at the site.



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
