# Peer review of "Three years of soil moisture observations by a dense cosmic-ray neutron sensing cluster at an agricultural research site in north-east Germany"

_Earth System Science Data, 2023_

## Author Comment (AC1)

**Author Response to Referee #1**

**Three years of soil moisture observations by a dense cosmic-ray neutron sensing cluster at an agricultural research site in north-east Germany**

Maik Heistermann et al.
*Earth Syst. Sci. Data Discuss.,* `doi:10.5194/essd-2023-19`
* * *
**RC:** *Referee Comment*,     AR: *Author Response*,     ☐ Manuscript text

Dear referee,

thank you very much for your positive response, and for the time and effort spent to examine the manuscript and the data set.

Your comments, even if minor, are very helpful. Please find a point-by-point reply below.

Kind regards,
Maik Heistermann (on behalf of the author team)

**Comments and responses**

**RC:** *[...] It would be good to see a volumetric soil moisture dataset for the static CRNS so that non-experts can use the data without having to derive their own calibration. The data is there for those with more experience to run their own calibration procedure.*

**AR:** We appreciate the suggestion, and understand the motivation behind it. Originally, we had decided *not* to include CRNS-based soil moisture estimates, $\theta(N)$, in this data publication since the results depend very much on the chosen processing methods, calibration strategies, temporal resolution etc.. Hence, in our view estimation products should not be part of such an observational data set. And, if we formally included $\theta(N)$ in the data publication, we would also have to comprehensively document the retrieval workflow in the manuscript which is, in our view, beyond the scope and the aims of the data paper.

Still, we agree that non-expert users should be enabled to make use of the data. Hence, we suggest a compromise: we will add, for each stationary CRNS location, the volumetric soil moisture $\theta(N)$ estimated from daily average neutron intensities (i.e. a daily soil moisture product), and briefly outline the main features of the underlying workflow in the meta data. In that context, we will also emphasize that the results are sensitive to the applied methodology, and that other users might come up with different values of $\theta(N)$. As we consider these additional files as a "convenience" feature, we would prefer not to extensively highlight and document this additional product in the data paper (manuscript) itself, except for a note at the end of the first paragraph of section 5, after l. 381 of the preprint:

> The data subsets for the stationary CRNS contain an exemplary soil moisture product retrieved from observed neutron counts.

Technically, we will create two directories in the subset "crns_stationary.zip": the directory "observations" will include the original stationary CRNS observations, as before. The directory "products" will contain the daily CRNS-based soil moisture product, including the corresponding meta data.

**RC:** *L14 – would be useful if this was a hyperlink. It is, after all, the main focus of the paper.*

AR: We fully agree. In the submitted version of the preprint, we included a hyperlink in the pdf, but apparently, it was removed in the course of preprint posting. In fact, all hyperlinks were removed. We will discuss this issue with the editorial office in case of an acceptance of the manuscript.

**RC:** *L24 – suggest "allowing users to obtain"*

AR: Will be implemented.

**RC:** *L57 – suggest "hence allowing"*

AR: Will be implemented.

**RC:** *L62 – suggest "...spanning all seasons and providing observations across more diverse..."*

AR: Will be implemented.

**RC:** *L66 – suggest " ...not only allows studying of the effect of the vertical soil moisture distribution on the CRNS signal, but also, together with groundwater observations, facilitates investigation of the processes of infiltration and groundwater recharge."*

AR: Will be implemented.

**RC:** *L71 – suggest "...muon-monitoring allows studying of novel methods for the local correction..."*

AR: Will be implemented.

**RC:** *Table 1, 3.4 – does the Roving data not include lat/long and elevation?*

AR: This is correct. We will include these variables in Tab. 1.

**RC:** *L103 – isn't the Quaesta instrument the same that is sold through Hydroinnova just under licence?*

AR: No, the instruments with IDs 11, 26, 27 and 30 (Tab. 2) are only provided by Quaesta, as far as we know.

**RC:** *L150 – how is this justified/calculated?*

AR: The purpose of providing an estimate of $N_0$ for the mobile CRNS in the manuscript was to give experienced users some guidance on how to obtain plausible soil moisture estimates from the roving observations. However, the value of $N_0$ typically depends on the retrieval method, which is a non-trivial procedure beyond the scope of this data paper: it includes the spatial distribution of the calibration measurements, the standardization of neutron intensities, and the accounting for the effects of incoming neutrons, humidity and barometric pressure, as well as soil organic matter, lattice water and vegetation.

Based on the referee comment, we think that it would indeed be more informative and consistent to provide – instead of an estimate of $N_0$ – a measure of the efficiency (i.e., sensitivity) of the roving CRNS in comparison

to the stationary CRNS (see also Table 2). That way, users themselves can decide how to estimate $N_0$ on the basis of the efficiency-corrected count rates and the featured dataset. Hence, we suggest to replace the sentence in lines 149-151 of the preprint,

> Since the sensor on a handwagon is not shielded by car material, we used a slightly larger calibration factor $N_0$ = 13447 cph compared to other studies.

by the following sentence:

> In relation to the calibrator CRNS (compare Tab. 2 and Sect. 3.2), the handwagon rover has a sensitivity of 5.51, i.e. in the same interval, it will count 5.51 times the number of epithermal neutrons detected by the calibrator probe (see also Heistermann et al., 2021).

**RC:** *L163 – what depth does the GNSS method reflect? Is it fixed or moisture dependent – more information is required*

AR: The penetration depth of GNSS-R in fact varies with soil moisture. More precisely, the variation of the apparent reflection depth is the key factor in soil moisture retrieval (see ll. 160-161). According to Larson et al. (2010), the penetration depth varies between 1 cm (wet soil) and 6 cm (very dry soil). We will revise the manuscript in order to include this information.

**RC:** *L217 – do you mean accuracy of 0.05 m3/m3?*

AR: Yes, correct... thanks for spotting this mishap!

**RC:** *L201 – suggest "... using a weighing pluviometer..."*

AR: Will be implemented.

**RC:** *Figure 6 caption – suggest "weight-based pluviometer" or just "weighing pluviometer"*

AR: Will be implemented.

**RC:** *Climate.zip – the files are .csv (comma separated values) but the text in the files is tab delimited. They should be .tsv or .tab files*

AR: We agree that this is an unnecessary source of confusion. We will revise all files and consistently use tab as separators, and replace the misleading extension ".csv" by ".txt".

**RC:** *CRNS_roving.zip – again the files are not comma delimited – they are tab delimited so the file extension is not correct*

AR: See previous response.

**RC:** *CRNS_stationary.zip – it would be nice to have an actual worked version of the soil moisture data with a time series of CRNS soil moisture – this would help non-CRNS researchers to use the data based on the processing method that you specify. All the other data for processing is alternative ways is there if required for more advanced users.*

AR: Please see our response to the first comment.

**RC:** *Soil_moisture_manual_sampling.zip – the file called theta_grav_OM_LW_texture.txt does not have lattice water values.*

AR: Sorry for the confusion. The file name was incorrect, it should be "theta_grav_OM_texture.txt" instead. Lattice water content was only determined for composite samples of two adjacent depths, so lattice water content for these composite samples can be found in "OM_LW_composite.txt". We will correct the file name and also update the file name in the meta data json-file.

**References**

Heistermann, M., Francke, T., Schrön, M., and Oswald, S. E. (2021): Spatio-temporal soil moisture retrieval at the catchment scale using a dense network of cosmic-ray neutron sensors, Hydrol. Earth Syst. Sci., 25, 4807–4824, https://doi.org/10.5194/hess-25-4807-2021.

Larson, K. M., J. J. Braun, E. E. Small, V. U. Zavorotny, E. D. Gutmann and A. L. Bilich (2010): GPS Multipath and Its Relation to Near-Surface Soil Moisture Content, IEEE Journal of Selected Topics in Applied Earth Observations and Remote Sensing, 3(1), 91-99, https://doi.org/10.1109/JSTARS.2009.2033612.

---

## Author Comment (AC2)

**Author Response to Referee #2**

**Three years of soil moisture observations by a dense cosmic-ray neutron sensing cluster at an agricultural research site in north-east Germany**

Maik Heistermann et al.

*Earth Syst. Sci. Data Discuss.,* `doi:10.5194/essd-2023-19`
* * *
**RC:** *Referee Comment*,     AR: *Author Response*,     ☐ Manuscript text

Dear referee,

thank you very much for your positive response, and for the time and effort spent to examine the manuscript and the data set. Please find our response to your comments below.

Kind regards,
Maik Heistermann (on behalf of the author team)

**Comments and responses**

**RC:** *[...] I suggest adding one column in Table 1 showing the range of measured values on this site with an indication of the mean or median value, for each measured variable.*

AR: We appreciate the suggestion. However, Tab. 1 is already very bulky, and represents a great diversity of variables (also for a large number of locations). Providing statistics like minimum, maximum, mean or median for all these variables in the context of such a table is, in our view, not feasible *in a meaningful way*, and also not in the interest of the audience. We would prefer not to extend Tab. 1 in such a way. For the key features of the dataset, we think that Fig. 3 provides an intuitive overview that should better serve the purpose. We hope that the referee agrees.

---

## Author Response (AR2)

**Author Response to Referees**

**Three years of soil moisture observations by a dense cosmic-ray neutron sensing cluster at an agricultural research site in north-east Germany**

Maik Heistermann et al.
*Earth Syst. Sci. Data Discuss.,* `doi:10.5194/essd-2023-19`
* * *
RC: *Referee Comment*,     AR: *Author Response*,     □ Manuscript text

Dear referees,

thank you very much for the positive responses, and for the time and effort spent to examine the manuscript and the data set.

With this letter, we provide the responses to both referee reports in one document. They basically correspond to our previous responses in the interactive discussion.

We hope that the revised versions of the manuscript and the dataset meet the standards of ESSD.

Kind regards,
Maik Heistermann (on behalf of the author team)

**1. Responses to referee #1**

**RC:** *[...] It would be good to see a volumetric soil moisture dataset for the static CRNS so that non-experts can use the data without having to derive their own calibration. The data is there for those with more experience to run their own calibration procedure.*

**AR:** We appreciate the suggestion, and understand the motivation behind it. Originally, we had decided *not* to include CRNS-based soil moisture estimates, $\theta(N)$, in this data publication since the results depend very much on the chosen processing methods, calibration strategies, temporal resolution etc. . In our view estimation products should not be part of such an observational data set. And, if we formally included $\theta(N)$ in the data publication, we would also have to comprehensively document the retrieval workflow in the manuscript which is, in our view, beyond the scope and the aims of the data paper.

Still, we agree that non-expert users should be enabled to make use of the data. Hence, we suggest a compromise: we will add, for each stationary CRNS location, the volumetric soil moisture $\theta(N)$ estimated from daily average neutron intensities (i.e. a daily soil moisture product), and outline the main features of the underlying workflow in the meta data. In the metadata, we will also emphasize that the results are sensitive to the applied methodology, and that other users might come up with different values of $\theta(N)$. As we consider

these additional files as a "convenience" feature, we would prefer not to extensively highlight and document this additional product in the data paper (manuscript) itself, except for a note at the end of the first paragraph of section 5, after l. 381 of the preprint:

> Furthermore, the data subset for the stationary CRNS (see Tab. 4) contains an exemplary daily soil moisture product retrieved from observed neutron counts.

Technically, we created two directories in the subset "crns_stationary.zip": the directory "observations" included the original stationary CRNS observations, as before. The directory "products" contains the daily CRNS-based soil moisture product, including the corresponding meta data. We also refer to this new subset of data in the README document (Tab. 1) at the upper level of the data repository.

**RC:** *L14 – would be useful if this was a hyperlink. It is, after all, the main focus of the paper.*

AR: We fully agree. In the submitted version of the preprint, we included a hyperlink in the pdf, but apparently, it was removed in the course of preprint posting. In fact, all hyperlinks were removed. We will try to make sure that the hyperlinks are not removed in the course of typesetting.

**RC:** *L24 – suggest "allowing users to obtain"*

AR: Implemented.

**RC:** *L57 – suggest "hence allowing"*

AR: Implemented.

**RC:** *L62 – suggest "...spanning all seasons and providing observations across more diverse..."*

AR: Implemented.

**RC:** *L66 – suggest " ...not only allows studying of the effect of the vertical soil moisture distribution on the CRNS signal, but also, together with groundwater observations, facilitates investigation of the processes of infiltration and groundwater recharge."*

AR: Implemented.

**RC:** *L71 – suggest "...muon-monitoring allows studying of novel methods for the local correction…"*

AR: Implemented.

**RC:** *Table 1, 3.4 – does the Roving data not include lat/long and elevation?*

AR: This is correct. We included these variables in Tab. 1.

**RC:** *L103 – isn't the Quaesta instrument the same that is sold through Hydroinnova just under licence?*

AR: No, the instruments with IDs 11, 26, 27 and 30 (Tab. 2) are only provided by Quaesta, as far as we know.

**RC:** *L150 – how is this justified/calculated?*

AR: The purpose of providing an estimate of $N_0$ for the mobile CRNS in the manuscript was to give experienced users some guidance on how to obtain plausible soil moisture estimates from the roving observations. However, the value of $N_0$ typically depends on the retrieval method, which is a non-trivial procedure beyond the scope of this data paper: it includes the spatial distribution of the calibration measurements, the standardization of

neutron intensities, and the accounting for the effects of incoming neutrons, humidity and barometric pressure, as well as soil organic matter, lattice water and vegetation.

Based on the referee comment, we think that it would indeed be more informative and consistent to provide – instead of an estimate of $N_0$ – a measure of the efficiency (i.e., sensitivity) of the roving CRNS in comparison to the stationary CRNS (see also Table 2). That way, users themselves can decide how to estimate $N_0$ on the basis of the efficiency-corrected count rates and the featured dataset. Hence, we replaced the sentence in lines 149-151 of the preprint,

> Since the sensor on a handwagon is not shielded by car material, we used a slightly larger calibration factor $N_0$ = 13447 cph compared to other studies.

by the following sentence:

> In relation to the calibrator CRNS (compare Tab. 2 and Sect. 3.2), the handwagon rover has a sensitivity of 5.51, i.e. in the same interval, it will count 5.51 times the number of epithermal neutrons detected by the calibrator probe (see also Heistermann et al., 2021).

**RC:** *L163 – what depth does the GNSS method reflect? Is it fixed or moisture dependent – more information is required*

AR: The penetration depth of GNSS-R in fact varies with soil moisture. More precisely, the variation of the apparent reflection depth is the key factor in soil moisture retrieval (see ll. 160-161). According to Larson et al. (2010), the penetration depth varies between 1 cm (wet soil) and 6 cm (very dry soil). We revised the manuscript in order to include this information in l. 162 of the original preprint, after "... assuming that other surface properties remain constant.":

> The corresponding penetration depth and hence representativeness of the signal amounts to around 1 cm for wet and up to 6 cm for very dry soils (Larson et al. 2010).

**RC:** *L217 – do you mean accuracy of 0.05 m3/m3?*

AR: Yes, correct... thanks for spotting this mishap!

**RC:** *L201 – suggest "... using a weighing pluviometer..."*

AR: Implemented.

**RC:** *Figure 6 caption – suggest "weight-based pluviometer" or just "weighing pluviometer"*

AR: Implemented.

**RC:** *Climate.zip – the files are .csv (comma separated values) but the text in the files is tab delimited. They should be .tsv or .tab files*

AR: We agree that this is an unnecessary source of confusion. We revised all files and consistently used tab as separators, and replaced the misleading extension ".csv" by ".txt".

**RC:** *CRNS_roving.zip – again the files are not comma delimited – they are tab delimited so the file extension is not correct*

 **AR:** See previous response.

**RC:** *CRNS_stationary.zip – it would be nice to have an actual worked version of the soil moisture data with a time series of CRNS soil moisture – this would help non-CRNS researchers to use the data based on the processing method that you specify. All the other data for processing is alternative ways is there if required for more advanced users.*

 **AR:** Please see our response to the first comment.

**RC:** *Soil_moisture_manual_sampling.zip – the file called theta_grav_OM_LW_texture.txt does not have lattice water values.*

 **AR:** Sorry for the confusion. The file name was incorrect, it should be "theta_grav_OM_texture.txt" instead. Lattice water content was only determined for composite samples of two adjacent depths, so lattice water content for these composite samples can be found in "OM_LW_composite.txt". We corrected the file name and also updated the file name in the meta data json-file.

**2. Response to referee #2**

**RC:** *[...] I suggest adding one column in Table 1 showing the range of measured values on this site with an indication of the mean or median value, for each measured variable.*

 **AR:** We appreciate the suggestion. However, Tab. 1 is already very bulky, and represents a great diversity of variables (also for a large number of locations). Providing statistics like minimum, maximum, mean or median for all these variables in the context of such a table is, in our view, not feasible *in a meaningful way*, and also not in the interest of the audience. We would prefer not to extend Tab. 1 in such a way. For the key features of the dataset, we think that Fig. 3 provides an intuitive overview that should better serve the purpose. We hope that the referee agrees.

**3. Other changes not related to referee comments**

We applied a few corrections which were not motivated by any referee comment.

- Tab. 2: in the column "Tubes", the entry "mod+bare" should be in the row with ID=27 instead of 26. ID=26 should have just "mod" instead. This was corrected.

- ll. 313-237 of the original preprint: based on a more recent analysis with regard to the allometric biomass estimation, we updated the manuscript text including the estimates for the average above-ground dry biomass for the two forest plots in the north and west and for the poplar plantation.

- The CRNS-based estimation of soil water content was slightly updated (inclusion of additional data for calibration, adjustment of offset resulting from measurement of soil organic matter and lattice water). This does not affect the dataset described in the manuscript, but introduced slight changes in those figures which include exemplary CRNS-based time series of SWC (Fig. 3-5).

- In Fig. 6, we included the most recent corrections of above-ground biomass, slightly changing the neutron count rates in panel b. The caption for Fig. 6b was adjusted accordingly ("b) corrected (pressure, humidity, incoming, biomass) epithermal neutron counts")

- In the data subset "Irrigation_experiment.zip", we corrected a mistake in the meta data file "meta.json" (replaced "northern" by "southern" in the description of the irrigation experiment).

- We updated the DOI to the latest version of the dataset in the manuscript (abstract, data availability section, references).

**References**

Heistermann, M., Francke, T., Schrön, M., and Oswald, S. E. (2021): Spatio-temporal soil moisture retrieval at the catchment scale using a dense network of cosmic-ray neutron sensors, Hydrol. Earth Syst. Sci., 25, 4807–4824, https://doi.org/10.5194/hess-25-4807-2021.

Larson, K. M., J. J. Braun, E. E. Small, V. U. Zavorotny, E. D. Gutmann and A. L. Bilich (2010): GPS Multipath and Its Relation to Near-Surface Soil Moisture Content, IEEE Journal of Selected Topics in Applied Earth Observations and Remote Sensing, 3(1), 91-99, https://doi.org/10.1109/JSTARS.2009.2033612.